# Assessment of the Shank-to-Vertical Angle While Changing Heel Heights Using a Single Inertial Measurement Unit in Individuals with Incomplete Spinal Cord Injury Wearing an Ankle-Foot-Orthosis

**DOI:** 10.3390/s21030985

**Published:** 2021-02-02

**Authors:** Lysanne A. F. de Jong, Yvette L. Kerkum, Tom de Groot, Marije Vos-van der Hulst, Ilse J. W. van Nes, Noel L. W. Keijsers

**Affiliations:** 1Department of Research, Sint Maartenskliniek, 6500 GM Nijmegen, The Netherlands; t.degroot@maartenskliniek.nl (T.d.G.); n.keijsers@maartenskliniek.nl (N.L.W.K.); 2Department of Rehabilitation, Donders Institute for Brain, Cognition and Behaviour, Radboud University Medical Center, 6525 AJ Nijmegen, The Netherlands; 3Research and Development, OIM Orthopedie, 9400 AE Assen, The Netherlands; Y.Kerkum@oim.nl; 4Department of Rehabilitation, Sint Maartenskliniek, 6500 GM Nijmegen, The Netherlands; M.Vos-vandeHulst@maartenskliniek.nl (M.V.-v.d.H.); I.vanNes@maartenskliniek.nl (I.J.W.v.N.)

**Keywords:** movement sensor, orthotics, AFO tuning, gait analysis, biomechanics, rehabilitation

## Abstract

Previous research showed that an Inertial Measurement Unit (IMU) on the anterior side of the shank can accurately measure the Shank-to-Vertical Angle (SVA), which is a clinically-used parameter to guide tuning of ankle-foot orthoses (AFOs). However, in this context it is specifically important that differences in the SVA are detected during the tuning process, i.e., when adjusting heel height. This study investigated the validity of the SVA as measured by an IMU and its responsiveness to changes in AFO-footwear combination (AFO-FC) heel height in persons with incomplete spinal cord injury (iSCI). Additionally, the effect of heel height on knee flexion-extension angle and internal moment was evaluated. Twelve persons with an iSCI walked with their own AFO-FC in three different conditions: (1) without a heel wedge (refHH), (2) with 5 mm heel wedge (lowHH) and (3) with 10 mm heel wedge (highHH). Walking was recorded by a single IMU on the anterior side of the shank and a 3D gait analysis (3DGA) simultaneously. To estimate validity, a paired t-test and intraclass correlation coefficient (ICC) between the SVA_IMU_ and SVA_3DGA_ were calculated for the refHH. A repeated measures ANOVA was performed to evaluate the differences between the heel heights. A good validity with a mean difference smaller than 1 and an ICC above 0.9 was found for the SVA during midstance phase and at midstance. Significant differences between the heel heights were found for changes in SVA_IMU_ (*p* = 0.036) and knee moment (*p* = 0.020) during the midstance phase and in SVA_IMU_ (*p* = 0.042) and SVA_3DGA_ (*p* = 0.006) at midstance. Post-hoc analysis revealed a significant difference between the ref and high heel height condition for the SVA_IMU_ (*p* = 0.005) and knee moment (*p* = 0.006) during the midstance phase and for the SVA_IMU_ (*p* = 0.010) and SVA_3DGA_ (*p* = 0.006) at the instant of midstance. The SVA measured with an IMU is valid and responsive to changing heel heights and equivalent to the gold standard 3DGA. The knee joint angle and knee joint moment showed concomitant changes compared to SVA as a result of changing heel height.

## 1. Introduction

Persons with an incomplete spinal cord injury (iSCI) experience sensory and motor deficits, such as muscle weakness, spasticity and impaired muscle coordination [1,2]. These motor deficits often result in an abnormal walking pattern and an increased fall incidence [1,2,3,4]. Although many persons with iSCI regain some walking capacity within the first six months, they often do not fully recover to their normal walking pattern [5]. To further improve walking in persons with iSCI, an ankle-foot orthosis (AFO) is often prescribed [6]. An AFO promotes normal joint kinematics and kinetics and improves spatiotemporal gait parameters and gait efficiency [7,8,9,10].

To optimize the effect of the AFO, the AFO footwear combination (AFO-FC) should be properly aligned [11]. Optimal alignment can be achieved by making fine adjustments to heel height, footplate stiffness and/or footwear, which is often referred to as tuning. The goal of AFO tuning is to align the ground reaction force (GRF) as close as possible to the knee joint center to minimize the knee flexion-extension moment [12]. In clinical practice, AFO tuning can be quantified by the shank-to-vertical angle (SVA), which is the angle between the anterior surface of the tibia and the vertical in the global sagittal plane [13]. When the shank is tilted posteriorly, the SVA is considered reclined, and the SVA is considered inclined when the shank is tilted anteriorly. An inclined SVA at midstance between 7° and 15°, with an optimum of 10° to 12°, facilitates appropriate orientation of the GRF in relation to the joint rotation centers [13]. Literature has shown that increasing the heel height of rigid AFO-FCs affected the sagittal knee angle and knee joint moment, which were reflected by an increase in the SVA [14].

Recent research has shown that an inertial measurement unit (IMU) on the anterior side of the shank is a valid and reliable instrument to measure the SVA in healthy individuals [15,16]. An advantage of this approach is that you do not require an expensive gait lab and time-consuming 3D gait analysis, which is considered as the gold standard, to measure the GRF in relation to the knee joint center. Therefore, IMUs are promising as an easy and low-cost alternative to guide AFO tuning. However, assessment of the SVA using an IMU also needs to be valid in individuals with neurological disorders wearing an AFO. Moreover, for AFO tuning it is specifically important that differences in the SVA measured with an IMU are detected during the tuning process, i.e., when adjusting heel height.

This study investigated the use of a single IMU on the shank for assessing the SVA at midstance in iSCI patients while wearing an AFO. The two main aims of this study were: (1) to examine the validity of an IMU to assess the SVA, and (2) to examine the responsiveness of the SVA measured with an IMU while changing the AFO-FC’s heel height. The SVA was measured by IMUs and 3D gait analysis (3DGA) as gold standard. As a secondary aim, the effect of heel height on the sagittal knee joint angle and knee joint moment were evaluated. It is hypothesized that an IMU is a valid method to assess the SVA in individuals with iSCI wearing an AFO and that the SVA measured with an IMU is responsive to changing heel heights and equivalent to 3DGA. The SVA measured with an IMU, as well as the SVA measured with 3DGA, knee joint angle and knee joint moment, are expected to increase during midstance with increasing heel height.

## 2. Materials and Methods

### 2.1. Participants

Persons with an iSCI who visited the Sint Maartenskliniek between June and December 2019 were screened for eligibility by a rehabilitation physician. Participants were included if they were (1) between 18–80 years old, (2) using an AFO with plantar flexion restriction, (3) using their AFO for a minimum of one month and (4) able to walk 10 m without a walking aid with and without AFO.

All participants gave written informed consent in accordance with the Declaration of Helsinki. The study was approved by the internal review board of the Sint Maartenskliniek and the regional medical ethics committee of Arnhem-Nijmegen (2018-4647).

Participant characteristics, like the American Spinal Injury Association (ASIA) Impairment Scale [17], level of the lesion, time since injury and Medical Research Council (MRC) scale for muscle strength [18] were extracted from the medical record.

### 2.2. Equipment

The study was conducted on a 10 m walkway in the gait laboratory of the Sint Maartenskliniek. This laboratory is equipped with a 10-camera motion capture system (Vicon, Oxford, MS, USA) and a force plate (Kistler Instruments, Hampshire, UK). One IMU (APDM, Portland, OR, USA) was placed on the shank. Data were recorded at 100 Hz for the motion capture system, 2400 Hz for the force plate and 128 Hz for the IMU. The three systems were synchronized with a trigger.

### 2.3. Measurement Procedure

Reflective markers were placed on the participant according to the Plug-in Gait lower body model. Three additional markers were attached to the shank to assess the SVA (Figure 1A). The IMU was placed on the anterior side of the shank with the vertical axis of the IMU aligned with the line connecting tibial tuberosity and midline of the ankle [15]. The researcher visually inspected if the IMU was aligned with the shank in a way the that anterior-posterior axis (Ximu) pointed in the walking direction. (Figure 1B). For each participant, the IMU was placed on the most affected leg with AFO. If the AFO consisted of an anterior shell, the IMU and the shank makers (SH1 and SH2) were placed on the AFO.

Participants performed a walking task with their own AFO-FC in three conditions, commonly used during the tuning process in clinical practice, in randomized order: (1) without a heel wedge (refHH), (2) with 5 mm heel wedge (lowHH) (Figure 1C) and (3) with 10 mm heel wedge (highHH) (Figure 1D). Participants were instructed to walk at a comfortable speed along the 10-m walkway. The measurement was completed when five successful trials were recorded (i.e., no irregular walking pattern observed, and a clean hit on the force plate of a single foot of the affected leg).

### 2.4. Data Analysis

The SVA measured with the IMU (SVA_IMU_) was calculated as described previously [15], using the quaternion data of the IMU. Quaternions were transformed into rotation matrices. The SVA_IMU_ was computed by the rotation of Z around Y, using the following equation:(1)$${\mathrm{SVA}}_{\mathrm{IMU}}=90-\mathrm{atan}\left(\frac{R_{{\mathrm{IMU}}_{1,3}}}{R_{{\mathrm{IMU}}_{3,3}}}\right)\times \frac{180}{\pi }$$
in which R_IMU_ is the rotation matrix of the IMU.

The SVA measured with the 3DGA (SVA_3DGA_) was calculated using the marker data. Marker data were filtered using the Woltring cross-validity quintic spline routine (MSE = 20) [19]. The SVA_3DGA_ was defined as the angle between the two markers at the anterior side of the shank (SH1 and SH2), and the vertical in the sagittal plane, and calculated using the following equation:(2)$${\mathrm{SVA}}_{3\mathrm{DGA}}=\mathrm{atan}\left(\frac{{\mathrm{posX}}_{\mathrm{SH}1}-{\mathrm{posX}}_{\mathrm{SH}2}}{{\mathrm{posZ}}_{\mathrm{SH}1}-{\mathrm{posZ}}_{\mathrm{SH}2}}\right)\times \frac{180}{\pi }$$
in which posX and posZ are the position of the shank markers on the anterior-posterior and vertical axis, respectively. Knee flexion-extension angles and internal moments were calculated using the Vicon Plug-In-Gait model and software.

Heel strike and toe-offs were calculated for the IMU using certain peaks in the angular velocity data during walking as has been previously described [15,20]. Position and acceleration of the foot markers were used to determine heel strikes and toe-offs for the 3DGA. Heel strike was defined as the mean of the instant that the vertical position of the heel marker was lowest and the heel marker maximally decelerated. Toe-off was defined as the mean of the instant that the vertical position of the toe marker increased and started to accelerate.

The SVA_IMU_, SVA_3DGA_, knee angle and knee moment were calculated at the instant of midstance, which was defined as defined as 50% between heel strike and toe-off. Because instants only reflect one specific point in time, instants could be more prone to error since a timing difference of a few samples affects the outcome, whereas the average during a phase will be less susceptible to this timing difference. Hence, we also calculated the average SVA_IMU_, SVA_3DGA_, knee angle and knee moment during the midstance phase, defined as 10–30% of the gait cycle [21]. To examine possible effects later on in the stance phase, average knee moment during terminal stance, defined as 30–50% of the gait cycle [21], was also calculated.

A timing difference between the instant of midstance was determined to explain possible differences between SVA_IMU_ and SVA_3DGA_. The difference in the instant of midstance in seconds between the IMU and 3DGA was calculated as the timing difference. Since kinematics and kinetics are influence by walking speed, walking speed for all heel height conditions was assessed. Walking speed was calculated as stride length divided by stride time based on 3DGA. All data processing and analyses were performed using MATLAB 2018b (The MathWorks Inc, Natick, MA, USA).

### 2.5. Statistical Analysis

In total 9 outcome measures were calculated, of which were 4 at midstance, 4 as the average during midstance phase and 1 at terminal stance. The SVA_IMU_, SVA_3DGA_, knee angle and knee moment at midstance and during midstance phase, and knee moment during terminal stance were averaged over five successful trials per condition. Participant characteristics were analyzed using descriptive statistics and presented as means ± standard deviations (SD).

Validity was estimated using a paired t-test and intraclass correlation coefficient (ICC) between the SVA_IMU_ and SVA_3DGA_ for the refHH. To estimate repeatability, the standard deviation for each participant across trials was calculated and averaged over all participants for the SVA_IMU_ and SVA_3DGA_.

For analyzing the responsiveness, a repeated measures ANOVA (α = 0.05) was conducted to examine differences in the SVA_IMU_, SVA_3DGA_, knee angle and knee moment between the heel height conditions (refHH, lowHH and highHH). Post hoc testing with Bonferroni correction (α = 0.05/3 = 0.0167) was performed to evaluate which conditions were significantly different from each other. Effect sizes (Cohen’s *d)* were calculated by dividing the mean difference between conditions by the standard deviation of the mean difference between conditions [22].

## 3. Results

### 3.1. Participants

Twelve persons with an iSCI (ten males/two females) with an average age ± SD of 55 ± 14 years were included in this study. More detailed participant characteristics are provided in Table 1. Due to technical issues with the motion capture cameras after moving the lab (e.g., missing markers), joint kinematics and kinetics could not be calculated accurately for three participants, who were therefore excluded from the knee angle and knee moment analyses.

### 3.2. Validity

The mean SVA_IMU_ was 11.3° ± 4.3° during midstance phase and 13.4° ± 4.2° at midstance, whereas the mean SVA_3DGA_ was 10.6° ± 4.2° during midstance phase and 13.6° ± 4.6° at midstance (Table 2). The mean ± SD difference between the SVA_IMU_ and SVA_3DGA_ for refHH during midstance phase was −0.69 ± 2.2 (t(11) = −1.10 *p* = 0.294) and 0.18 ± 2.6 (t(11) = 0.247, *p* = 0.809) at midstance. The ICCs were 0.93 and 0.91 for the midstance phase and the instant of midstance, respectively. The standard deviation across trails over the participants was 2.2° ± 1.2° for the SVA_IMU_ and 1.5° ± 1.6° for the SVA_3DGA_ during the midstance phase, and 2.1° ± 1.2° for the SVA_IMU_ and 0.95° ± 0.47° for the SVA_3DGA_ at midstance. The instant of midstance was 0.011 ± 0.014 (t(11) = 2.71, *p* = 0.020) seconds earlier for the IMU compared to 3DGA for the refHH.

### 3.3. Heel Height Conditions

Mean and standard deviations in SVA_IMU_, SVA_3DGA_, knee angle and knee moments for the heel height conditions are shown in Table 2. The mean SVA_IMU_, SVA_3DGA_, knee angle and knee moment during the gait cycle are shown in Figure 2. No statistical difference in walking speed was found between the heel height conditions. The repeated measures ANOVA revealed a significant main effect of heel height for the SVA_IMU_ (F(11,2) = 3.87, *p* = 0.036) and knee moment (F(8,2) = 5.03, *p* = 0.020) during the midstance phase (see Table 3 for differences between conditions). At midstance, SVA_IMU_ (F(11,2) = 3.68, *p* = 0.042) and SVA_3DGA_ (F(11,2) = 6.51, *p* = 0.006) were significantly different between the heel heights. Post-hoc testing showed a significant difference between the refHH and highHH for the SVA_IMU_ (*p* = 0.005) and knee moment (*p* = 0.006) during the midstance phase and for the SVA_IMU_ (*p* = 0.010) and SVA_3DGA_ (*p* = 0.006) at the instant of midstance. No significant differences for the other comparisons (refHH-lowHH and lowHH-highHH) were found. No main effect of heel height was found for SVA_3DGA_, knee angle during the midstance phase, for the knee angle and knee moment at the instant of midstance and knee moment during terminal stance.

## 4. Discussion

This study investigated the use of a single IMU to assess the SVA while changing heel heights in persons with iSCI. The validity was good with small mean differences and high ICCs above 0.9. The SVA increased as a result of increasing heel heights, which was assessed by the IMU and 3DGA. Additionally, we found that an increase in heel height resulted in concomitant changes in knee joint moment at midstance. Knee joint angle did not reach statistical significance as a result of increasing heel height.

The small difference and standard deviation together with the high ICCs in SVA between IMU and 3DGA supports the validity of assessing the SVA with an IMU as has shown before [15,16]. Likewise, the standard deviation of 2° across trials corresponded to the intra-session standard deviation of the SVA measured with a smartphone [16]. The timing difference in midstance of 0.01 s between IMU and 3DGA was comparable to the timing difference in healthy controls, indicating the IMU is able to determine midstance accurately [15].

The significant main effect of heel height for the SVA indicated responsiveness of the SVA to changes in AFO-FC heel height. In contrast to previous study with high heel height differences above 10 mm [14], a strength of our study is that we tested the participants with heel height conditions of 5 and 10 mm in accordance to clinical practice. The SVA measured by the IMU and 3DGA was only significantly different between the reference and high heel height condition in the post-hoc analysis. Since no differences of the low heel height with the reference and high heel height condition were found in the SVA, our results indicate that subtle changes in heel height are not reflected by the SVA in this study population. We found a larger mean and standard deviation for the differences in SVA between the heel heights measured with the IMU compared to 3DGA. The effect sizes, ranging from 0.26 to 1.03 (Table 3), were nearly similar indicating that the responsiveness of the SVA measured by an IMU is equivalent to the gold standard 3DGA.

In the current study, we also assessed the kinetics and kinematics of the lower limb to evaluate the alignment of an AFO. The knee joint moment at midstance was significantly different between conditions whereas the knee joint angle was nearly significant, indicating both parameters were influenced by AFO-FC heel height. This is in line with previous literature which found an increase in knee flexion angle and internal knee extensor moment as a result of an increased heel height [12,14,23,24]. Remarkably, the SVA had the same effect size as the knee joint moment between the heel height conditions (see Table 3), indicating that the responsiveness of the SVA corresponds to the responsiveness of the knee joint moment.

We evaluated the outcome measures SVA, knee angle and knee moment during midstance phase and at midstance. Because comparable results were found for all outcome measures, we prefer to use the outcome measures at midstance since the instant midstance can be easily detected when using IMUs and/or video. Furthermore, the instant of midstance corresponds to the clinically used definition of midstance, i.e., the instant when the swing leg passes the stance leg in the walking direction.

The inability to measure subtle changes, especially between the low and high heel height condition, could be explained by the great variability in response on increasing heel heights. Moreover, the AFOs were already tuned in clinical practice. As a result, participants may have counteracted the increase to prevent larger knee angles and moments in order to retain stability during stance and an efficient walking pattern. The ability to counteract the changing heel heights could be due to the familiarity of the participants to walk with their AFOs and/or their walking ability. Another explanation of the small differences could be the use of small heel height differences. Previous literature on AFO tuning using heel height adjustments used higher heel heights to increase the SVA [14]. Accordingly, this study found larger differences in SVA between conditions. However, these large heel heights do not reflect clinical practice. The great variability in the SVA measured by the IMU can be due to the movement of the IMU during and between trials. The IMU was attached to the shank using an elastic band around the calf muscle. Since most participants walked with an anterior supported AFO, there could be some movement between the shank and carbon fiber shell of the AFO, pulling on the elastic band, causing the IMU to slide down or sideways. We recommend attaching the IMU with double side tape to prevent movement of the IMU. The use of an anterior supported AFO resulted also in the attachment of the shank markers on the AFO instead of on the skin. It could be possible that the shank itself was not in contact with the anterior shell, resulting in a different orientation of the shank and AFO. However, due to the working mechanism of these AFOs and the inclusion of subjects with individually fitted AFOs, we believe that the shank is pushed against the anterior shell during midstance. Hence, the orientation of the shank will be similar to the orientation of the anterior shell, resulting in a correct measurement of the SVA with the shank markers on the AFO. Another explanation that needs to be addressed is the small sample size. Only 12 persons with iSCI participated in this study, and only 9 were included in the analyses for the knee angle and knee moment. This sample size was too low to find a main effect for knee angle and differences between the reference and low heel height. For clinical applications, however, the effect size should be large enough to reveal significant differences in such a small sample size.

Although the SVA is an important measure in clinical practice, and differences between heel height conditions can be measured [12,14,23,24], evidence for the relationship between the SVA and optimal AFO alignment is still scarce. The mean SVA of 11 degrees during midstance phase with the reference heel height (see Table 2) does support the idea that there is an optimum in SVA between 10 and 12 degrees [13]. However, the standard deviation of 4 degrees indicates that individuals deviate from this optimum. Therefore, adding an extra outcome parameter to the SVA could give more information on the optimal alignment of the AFO. The effect size between the reference and low heel height was highest for the knee angle. The knee angle can be easily measured by attaching an IMU to the thigh in addition to the shank for the SVA. Measuring SVA and knee angle during AFO tuning in a large population is needed to increase the understanding of optimal alignment of an AFO.

## 5. Conclusions

The SVA measured with an IMU is valid and responsive to changing AFO-FC heel height and is equivalent to the gold standard 3DGA. The knee joint angle and knee joint moment showed corresponding changes indicating that the SVA reflects changes in AFO alignment.

## Figures and Tables

**Figure 1 sensors-21-00985-f001:**
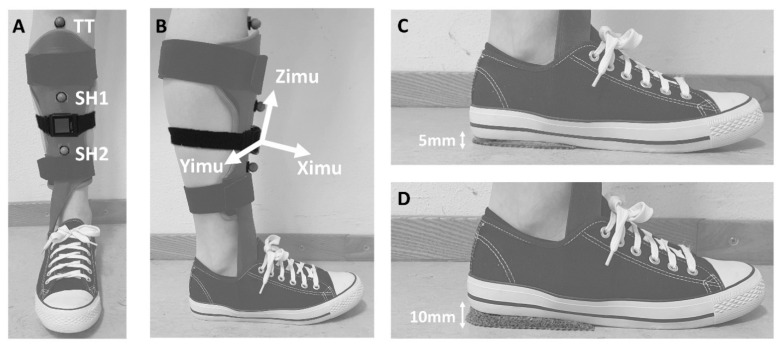
(**A**) Frontal view of the lower leg with additional markers on tibia tuberosity (TT) and the shank (SH1 and SH2). (**B**) Sagittal view of the lower leg with the corresponding coordinate system of the IMU (XYZimu). (**C**) Sagittal view of 5 mm heel wedge (lowHH). (**D**) Sagittal view of 10 mm heel wedge (highHH).

**Figure 2 sensors-21-00985-f002:**
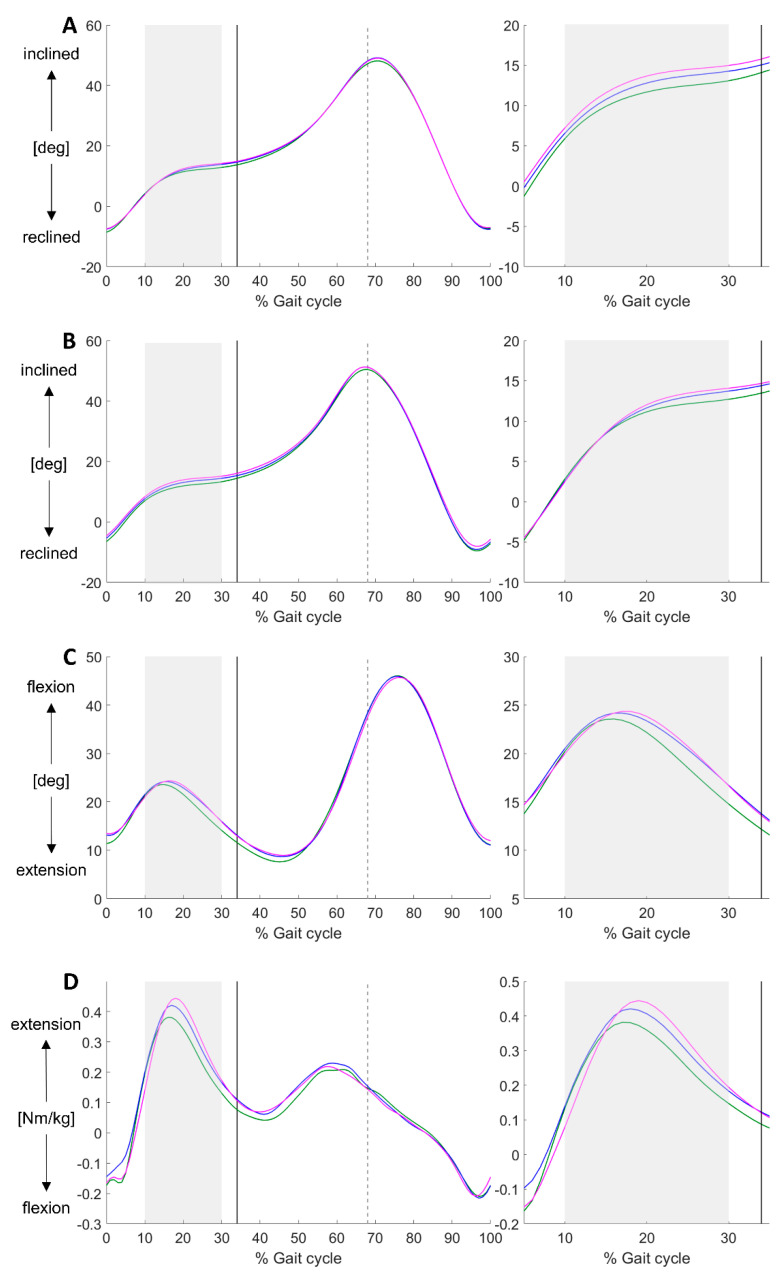
Mean SVA_IMU_ [deg] (**A**), SVA_3DGA_ [deg] (**B**), knee flexion-extension angle [deg] (**C**) and internal knee flexion-extension moment [Nm/kg] (**D**) during the gait cycle for refHH (green), lowHH (blue) and highHH (pink). The shaded area indicates the midstance phase (10–30%), the black line indicates the instant of midstance (34%) and the dashed line indicates toe off (68%).

**Table 1 sensors-21-00985-t001:** Participant characteristics (*n* = 12).

Characteristics	Mean ± SD
Age (years)	55.0 ± 14.1
Gender, male/female	10/2
Height (cm)	182.4 ± 8.7
Weight (kg)	90.0 ± 19.9
ASIA Impairment Scale, grade C/D	1/11
Level of the lesion, C1-8/T1-12/L1-5	4/4/4
Time since injury	161 ± 212
Affected leg, left/right	6/6
MRC plantar flexors, 0/1/2/3/4/5	3/1/2/4/1/1
MRC dorsal flexors, 0/1/2/3/4/5	1/2/0/5/3/1
AFO type, dynamic */hinged **	11/1

* dynamic anterior shell carbon fiber AFO; ** spring-hinged AFO with anterior shell.

**Table 2 sensors-21-00985-t002:** Mean ± SD values of the SVA_IMU_ (*n* = 12), SVA_3DGA_ (*n* = 12), knee flexion angle (*n* = 9) and internal knee flexion moment (*n* = 9) during midstance phase and at midstance, and internal knee flexion moment (*n* = 9) during terminal stance for the different heel height conditions.

Conditions	RefHH	LowHH	HighHH	F	*p*
Midstance phase					
SVA_IMU_ (°)	11.3 ± 4.3	12.4 ± 4.8	13.2 ± 4.3	3.87 ^a^	0.036
SVA_3DGA_ (°)	10.6 ± 4.2	10.9 ± 4.3	11.2 ± 4.2	2.51 ^a^	0.104
Knee angle (°)	20.2 ± 4.3	21.5 ± 3.8	21.6 ± 3.1	3.56 ^b^	0.053
Knee moment (Nm/kg)	0.28 ± 0.12	0.31 ± 0.12	0.33 ± 0.12	5.03 ^b^	0.020
Instant of midstance					
SVA_IMU_ (°)	13.4 ± 4.2	14.6 ± 4.5	15.3 ± 3.5	3.68 ^a^	0.042
SVA_3DGA_ (°)	13.6 ± 4.6	14.5 ± 4.3	14.8 ± 4.3	6.51 ^a^	0.006
Knee angle (°)	11.4 ± 6.3	12.7 ± 5.3	12.8 ± 5.2	3.03 ^b^	0.077
Knee moment (Nm/kg)	0.07 ± 0.16	0.11 ± 0.13	0.12 ± 0.14	3.27 ^b^	0.065
Terminal stance phase					
Knee moment (Nm/kg)	0.07 ± 0.16	0.10 ± 0.16	0.11 ± 0.15	2.24 ^b^	0.139
Gait characteristics					
Walking speed (m/s)	0.85 ± 0.18	0.84 ± 0.17	0.83 ± 0.17	1.66 ^a^	0.214

^a^ degrees of freedom: 11.2; ^b^ degrees of freedom: 8.2.

**Table 3 sensors-21-00985-t003:** Mean ± SD differences and effect sizes (ES) between the heel height condition of SVA_IMU_, SVA_3DGA_, knee angle and knee moment during midstance phase and at midstance, and knee moment during terminal stance.

Conditions	Ref–Low		Low–High		Ref–High	
	Mean ± SD	ES	Mean ± SD	ES	Mean ± SD	ES
Midstance phase						
SVA_imu_ (°)	1.1 ± 2.6	0.40	0.80 ± 2.4	0.33	1.9 ± 1.8 *	1.03
SVA_3DGA_ (°)	0.32 ± 1.1	0.28	0.27 ± 0.6	0.44	0.59 ± 0.9	0.65
Knee angle (°)	1.3 ± 1.8	0.69	0.16 ± 1.6	0.10	1.4 ± 1.9	0.76
Knee moment (Nm/kg)	0.04 ±0.07	0.54	0.02 ± 0.05	0.39	0.05 ± 0.04 *	1.43
Instant of midstance						
SVA_imu_ (°)	1.2 ± 2.7	0.44	0.6 ± 2.3	0.28	1.8 ± 2.0 *	0.89
SVA_3DGA_ (°)	0.9 ± 1.4	0.69	0.3 ± 1.0	0.26	1.2 ± 1.2 *	0.98
Knee angle (°)	1.3 ± 1.9	0.70	0.04 ± 1.5	0.03	1.4 ± 2.3	0.61
Knee moment (Nm/kg)	0.04 ± 0.08	0.50	0.01 ± 0.04	0.23	0.05 ± 0.06	0.84
Terminal stance phase						
Knee moment (Nm/kg)	0.03 ± 0.07	0.13	0.01 ± 0.04	0.08	0.04 ± 0.05	0.17

* Significant differences between the conditions (*p* < 0.0167).

## Data Availability

The data presented in this study are available on request from the corresponding author.

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
