# Peer review of "Assessment of the Shank-to-Vertical Angle While Changing Heel Heights Using a Single Inertial Measurement Unit in Individuals with Incomplete Spinal Cord Injury Wearing an Ankle-Foot-Orthosis"

_sensors, 2021, doi:10.3390/s21030985_

Round 1

Reviewer 1 Report

It seems to me a very interesting study and with great applicability to practice and to this type of patients
It would be interesting if it could be done with a larger number of participants.
I think more characteristics of the participants could be provided.

What kind of affectation do the participants have, it would be important to specify this because the results will not be the same

Table 1 seems too scarce to me, more variables could be added such as type of injury, affectation.

Musculature should be assessed

Author Response

We would like to thank the reviewers for their valuable time and considerate feedback. Please see the attachment for the point-to-point reply. 

Reviewer 2 Report

Comments to authors:

The paper describes the use of a single IMU, placed on the anterior side of the shank, to measure the shank-to-vertical angle (SVA). The authors also investigate the effect of heel heights on SVA (measured by the IMU), knee angle, and knee moment in persons with incomplete spinal cord injuries. Experimental data was collected from real people in hospital/clinic environment, and then it was treated statistically to extract meaningful information.

The work is valuable and useful for the people who works in the area. The paper is very “clean”: there is a problem, the authors propose a solution, the solution is tested experimentally, experimental data is analyzed scientifically, and conclusions are extracted. The work is not outstanding in terms of innovation (many teams have been using IMUs to study gait) but it is fair and solid.

The article is well organized, the text is well written, symbols and tables are consistent, and figures look good (in general). The overall aspect of the paper is good.

Experimental tests were done with real people, mixing knowledge from medicine, engineering, and data analytics. The dimension and diversity of the population is, in my opinion, acceptable.

Conclusion:  In my opinion, the paper, as it is, meets the requirements to be published in the Sensors MDPI Journal.

Author Response

(The authors gave the same response as above.)

Reviewer 3 Report

Thank you for the opportunity to review your article.

This manuscript reports the findings of a study to examine changes in shank-to-vertical angle (SVA) with different heel wedge heights in ankle foot orthosis (AFO) users. SVA is used to adjust AFOs during the fitting and tuning process, but existing methods to quantify SVA are not accessible to the orthotic clinic. Therefore, the authors tested a previously developed inertial measurement unit (IMU) method in a mock AFO tuning paradigm to determine if these wearable sensors could be useful to tune AFOs outside of a lab setting. The study found that differences in SVA as measured by IMU and traditional motion capture could be detected with a 10mm but not 5mm heel height difference from AFO users’ typical shoes and AFO condition. The authors concluded that IMUs can detect changes in SVA.

The manuscript is generally well-written and the rationale for the study as laid out in the introduction seems sound. My largest criticism with the study is that the introduction suggests that accurate measurement of SVA itself is important but the study does not report ability to accurately measure SVA in a population wearing AFOs. The need to reliably detect a change in SVA is supported, but it would seem that detecting a change in SVA is only useful if you know SVA, since there is a recommended range of SVA for AFO tuning. Based on the results (but not the methods), it appears that the authors measured SVA and compared SVA itself between IMU and 3D motion capture measurement techniques. Mentioning this in the aims, methods, and discussion would add context to the examination of “responsiveness”.

Title

  1. The title doesn’t seem to align with the study. The study doesn’t assess AFO alignment, but rather how SVA changes with heel height in AFO users.

Keywords

  1. I suggest not repeating words from the title in the keywords. Using different words will increase the appearance of an article in search results.

Introduction

  1. Lines 47-49: Please define shank to vertical angle more specifically. Do larger angles indicate anterior or posterior rotation of the proximal relative to distal shank (i.e., anteriorly or posteriorly tilted shank)? Also, for clarification, does “shank” in this context refer to the anterior surface of the lower leg (as it seems based on Figure 1 and reference 15)? This would be different than the common biomechanics use of “shank” as a reference to the entire lower leg/tibia.

  1. If the specific angle of the shank is important for AFO tuning, why test for the ability to detect differences in SVA between shoe conditions without reporting the actual SVA for the IMU and 3DGA across conditions? Especially after reading the results, I suggest rewording the aims.

Materials and Methods

  1. Line 73: Does “an AFO with plantar flexion restriction” include multiple types of AFO? Can this be more specific about AFO types included and excluded? E.g., solid, posterior leaf spring, hinged?

  1. Lines 87-92: Figure 1 suggests that the shank IMU and markers were at least sometimes placed on the AFO. Was this the case in all participants or were the IMU and markers placed on the skin for some participants? If markers and IMU are placed on the AFO, the measurement is not necessarily SVA, it’s AFO-VA.

  1. What is the rationale for the heel wedge heights?

  1. Was the heel height of participants’ shoes recorded? I’d assume most shoes had a larger heel-to-toe drop than the shoes pictured in Figure 1. Could there have been an interaction between shoe heel height and response of SVA to the experimental conditions?

  1. Was walking speed measured or controlled within or between conditions?

  1. Was magnetometer data included in quaternion calculations?

  1. Lines 117-118: What was the difference in heel strike and toe-off identification times between the IMU and marker methods? Why were different gait event definitions used for these synchronized data types? Differences in gait event identification could lead to the appearance of differences in outcome measures.

  1. Lines 123-124: What is the definition of the midstance phase (10-30% of the gait cycle) based on? This does not seem to be the middle portion of stance. In Figure 2, it appears that the instant of midstance is outside of the midstance phase which is confusing.

  1. Line 123: Was the knee moment at the instant of midstance always a flexion (positive) moment? If not, it may be more appropriate to compare the mean of the absolute value of the moment at this instant across participants instead of the mean of the true value of this moment if the goal of AFO alignment is to reduce the absolute magnitude of the knee moment.

  1. Lines 124-128: I am unsure of the interpretability of some of the outcome variables when they are averaged over large chunks of the gait cycle. The knee angle and knee moment in particular vary greatly across the “midstance phase”. I’m not sure how an average over the phase meaningfully represents these data.

  1. Line 127: “stance phase” rather than “stand phase”.

  1. Lines 137-139: Were these one-way repeated measures ANOVAs by condition only? The results reporting suggests that some statistical test was performed to compare between IMU and 3DGA as well but I do not see a test here that aligns with that.

Results

  1. Table 1: “Height” rather than “Length”?

  1. Table 1: Define ASIA grade and include how this was assessed in the methods.

  1. Lines 152-157: How were the data in this section calculated? Further, no statistical analyses are mentioned in the methods for the comparisons made in this section.

  1. Figure 2: Are these mean data across all subjects? Suggest including standard deviation.

  1. Figure 2: I cannot see a dotted line in these figures.

  1. Figure 2: The y axes are unclear. Can the labels be rotated to more clearly indicate which ends of the axes correspond to flexion/extension or reclined/inclined. Also, please define what reclined and inclined mean in reference to shank angle.

Discussion

  1. I am unsure of the use of the current results in actual practice. Ability to detect differences in SVA is surely an important aspect of this measurement, but based on the introduction, I assumed that accurate measurement of SVA itself was the more important goal. How do these results advance the science towards this goal or, alternatively, why is the ability to detect the changes reported in this study useful?

  1. I suggest discussing the results related to the study hypotheses before discussion of other study considerations. Here, discussion of the differences in SVA between heel height conditions does not come until part way through the third paragraph.

  1. Lines 192-199: Some of this information would be useful in the methods. Without this rationale, I did not know why outcomes were calculated over these two separate time windows.

  1. Line 228: “large heel heights do not reflect clinical practice”. I believe the authors missed an opportunity to promote one of the strengths of their study design here. By my read of this section, the current method can detect differences in SVA between clinically relevant 10mm differences in heel height, where previous studies only demonstrated that their method could detect differences in SVA when differences in heel height were unrealistically large. If I am interpreting this correctly, I suggest moving this point earlier in the discussion. The limitation that 5mm differences couldn’t be detected is still relevant to mention.

  1. Lines 244-249: The data don’t support these statements. The ES for the knee angle was large for the 5mm difference between reference and lowHH, but not for the 5mm difference between lowHH and highHH.

  1. Line 248: “knee angle” not “knee ankle”.

Author Response

We would like to thank the reviewers for their valuable time and considerate feedback. Please find the attachment for the point-to-point reply. 

Round 2

Reviewer 3 Report

Thank you for the opportunity to review your revised manuscript.

Overall, the manuscript is much improved after revision. The extension of previous work demonstrating validity in healthy individuals to validity and responsiveness in a clinical population is clear. Rationale for specific analysis choices is improved. I have only a few minor suggestions remaining.

  1. I am still unsure what the ASIA grade is or what “1/11” in Table 1 means. Also in Table 1, what does “MRC” stand for? Please include information on where these characteristics were obtained (e.g., from medical records, as suggested in the review response, assessed during study visit, etc.).

  1. Section 3.2: Add units to results (lines 181-184, 186-187).

  1. Figure 2: The new light grey line is still very difficult to see. Image quality may be being altered in conversion to publication format because the “black” line shows up as a medium grey and the light grey line is barely visible.

  1. Line 274: Delete either “We evaluated” or “were evaluated.”

Author Response

We thank the reviewer for this considered time and feedback. Please see the attachment for th point-to-point reply.
